

# MK8617 inhibits M1 macrophage polarization and inflammation *via* the HIF-1α/GYS1/UDPG/P2Y$_{14}$ pathway

Lingling Qian[1,*], Xiao-qin Chen[2,*], Deyang Kong[3], Gaoyuan Wang[1], Yun Cao[2], Yingchun Xiao[2], Jing-yuan Cao[2] and Chunbo Zou[2]

[1] Department of Nephrology, Nanjing University of Chinese Medicine, Nanjing, Jiangsu, China
[2] Department of Nephrology, Taizhou School of Clinical Medicine, The Affiliated Taizhou People's Hospital of Nanjing Medical University, Taizhou, Jiangsu, China
[3] Department of Nephrology, Shenzhen Bao'an District Songgang People's Hospital, Shenzhen, Guangdong, China
* These authors contributed equally to this work.

## ABSTRACT

**Background:** Nonresolving inflammation is a major driver of disease and needs to be taken seriously. Hypoxia-inducible factor (HIF) is closely associated with inflammation. Hypoxia-inducible factor-prolyl hydroxylase inhibitors (HIF-PHIs), as stabilizers of HIF, have recently been reported to have the ability to block inflammation. We used MK8617, a novel HIF-PHI, to study its effect on macrophage inflammation and to explore its possible mechanisms.

**Methods:** Cell viability after MK8617 and lipopolysaccharide (LPS) addition was assessed by Cell Counting Kit-8 (CCK8) to find the appropriate drug concentration. MK8617 pretreated or unpretreated cells were then stimulated with LPS to induce macrophage polarization and inflammation. Inflammatory indicators in cells were assessed by real-time quantitative reverse-transcription polymerase chain reaction (qRT-PCR), western blot (WB) and immunofluorescence (IF). The level of uridine diphosphate glucose (UDPG) in the cell supernatant was measured by ELISA. Purinergic G protein-coupled receptor P2Y$_{14}$, as well as hypoxia-inducible factor-1α (HIF-1α) and glycogen synthase 1 (GYS1) were detected by qRT-PCR and WB. After UDPG inhibition with glycogen phosphorylase inhibitor (GPI) or knockdown of HIF-1α and GYS1 with lentivirus, P2Y$_{14}$ and inflammatory indexes of macrophages were detected by qRT-PCR and WB.

**Results:** MK8617 reduced LPS-induced release of pro-inflammatory factors as well as UDPG secretion and P2Y$_{14}$ expression. UDPG upregulated P2Y$_{14}$ and inflammatory indicators, while inhibition of UDPG suppressed LPS-induced inflammation. In addition, HIF-1α directly regulated GYS1, which encoded glycogen synthase, an enzyme that mediated the synthesis of glycogen by UDPG, thereby affecting UDPG secretion. Knockdown of HIF-1α and GYS1 disrupted the anti-inflammatory effect of MK8617.

**Conclusions:** Our study demonstrated the role of MK8617 in macrophage inflammation and revealed that its mechanism of action may be related to the HIF-1α/GYS1/UDPG/P2Y$_{14}$ pathway, providing new therapeutic ideas for the study of inflammation.

Corresponding authors
Jing-yuan Cao,
caojingyuan1989@163.com
Chunbo Zou, shilz123@yeah.net

## INTRODUCTION

Inflammation is the immune system's response to harmful stimuli, such as ischemia, infections or trauma (*Chen et al., 2017*). In such cases, various endogenous signals that normally reside inside cells, known as danger-associated molecular patterns (DAMPs), are released to the extracellular space. Once outside the cell, DAMPs are recognized by cellular receptors such as pattern-recognition receptors (PRRs), which then activate inflammatory pathways and recruit inflammatory cells (*Murao et al., 2021*). Macrophage recruitment occurs in the early stages of inflammation and it rapidly produces large amounts of inflammatory cytokines in response to danger signals, which can be considered as the trigger of the inflammatory response (*Ross, Devitt & Johnson, 2021*). Inflammation is a double-edged sword. Inflammation itself is a protective response to the elimination of harmful substances, but excessive inflammation can lead to destructive effects and drive the occurrence of diseases (*Kurlansky, 2015*). According to research, in addition to inflammatory diseases and autoimmune diseases, inflammatory responses are also the "common soil" for multifactorial diseases such as diabetes, obesity, asthma and cancer (*Scrivo et al., 2011*). Therefore, attaining an early response to macrophage inflammation and, thus, delaying disease progression are important therapeutic strategies (*Liu et al., 2014*; *Na et al., 2019*).

Hypoxia-inducible factor (HIF) is a nuclear transcription factor that is activated under hypoxic conditions, which comprises a functional α subunit and a structural β subunit (*Jaakkola et al., 2001*). It can orchestrate a metabolic switch that allows cells to survive in a hypoxic environment. Research shows that infected and inflamed tissues are often hypoxic, and HIF helps immune cells adapt (*McGettrick & O'Neill, 2020*). Therefore, targeting HIF may be a new idea to control inflammation. Hypoxia-inducible factor-prolyl hydroxylase inhibitors (HIF-PHIs) are novel therapeutic agents for renal anemia, found on the basis of HIF, which can stabilize HIF and prevent its degradation (*Gupta & Wish, 2017*). In addition, based on the role of HIF in immunity and inflammation, some studies have found that HIF-PHIs can also alleviate inflammation, thereby preventing various diseases (*Miao et al., 2021*; *Han et al., 2020*; *Kim et al., 2021*). HIF-PHIs have the potential to be novel agents for controlling inflammation, but the mechanisms are unclear and may be related to intracellular metabolism, especially glycogen metabolism (*McGettrick & O'Neill, 2020*; *Ito et al., 2020*).

Uridine diphosphate glucose (UDPG), a nucleotide sugar, is an intermediate metabolite of glycogen synthesis (*Adeva-Andany et al., 2016*). In addition, it also belongs to DAMPs, which can be secreted by cells to bind to the purinergic G protein-coupled receptor $P2Y_{14}$ on the cell membrane, thereby initiating onset of the downstream immune response (*Thwe et al., 2017*). The process of synthesizing glycogen from UDPG is mediated by glycogen synthase 1 (GYS1), which can be directly regulated by HIF-1α (*Pescador et al., 2010*).

Therefore, we hypothesized that the regulation of GYS1 by HIF-1α interferes with the secretion of UDPG and thus affects the inflammatory response, which may be the anti-inflammatory mechanism of HIF-PHIs. In our study, RAW264.7 macrophages were polarized to the M1 phenotype by exposure to lipopolysaccharide (LPS) to mimic the inflammatory response. We used MK8617, a recently discovered selective, orally-active HIF-PHI, to observe whether HIF-PHI pretreatment prevented inflammation (*Debenham et al., 2016*). It has been shown that high doses of MK8617 promote muscle inflammation in mice with chronic kidney disease, while the effects of low and moderate doses of MK8617 on inflammation were not mentioned (*Qian et al., 2019*). Therefore, here we tried to explore whether safe doses of MK8617 play a positive role in inflammation as well as other HIF-PHIs. Furthermore, we tried to explore the anti-inflammatory mechanisms of action of HIF-PHIs, focusing on whether rate-limiting enzymes and intermediates of glycogen are involved in suppressing the macrophage inflammatory response.

## MATERIALS AND METHODS

### Cell culture and treatment

RAW264.7 cells, a murine macrophage cell line, were purchased from the Chinese Academy of Medical Science (Shanghai, China). The cells were grown in Dulbecco's modified Eagle medium (DMEM; GibcoTM; ThermoFisher Scientific, Waltham, MA, USA) supplemented with 10% fetal bovine serum (FBS; AusGeneX, Molendinar, Queensland, Australia) and 1% penicillin-streptomycin (Servicebio, Wuhan, China). RAW264.7 cells were pretreated with different concentrations of MK8617 (Selleck, Houston, TX, USA) dissolved in 0.1% dimethyl sulfoxide (DMSO; Sigma-Aldrich, St. Louis, MO, USA) for 24 h, followed by 1μg/mL LPS (Sigma-Aldrich, St. Louis, MO, USA) dissolved in phosphate-buffered saline (PBS; Servicebio, Wuhan, China) for 24 h to induce into M1 phenotype. Cells were treated with equivalent amounts of vehicle (DMSO or PBS) as control. The post-treatment of cells with MK8617 was LPS treatment of cells for 24 h followed by MK8617 treatment for 24 h. The time of treating cells with MK8617 or LPS alone in the experiment was also chosen to be 24 h. The duration of UDPG (Solarbio® Life Sciences, Beijing, China) treatment of RAW264.7 cells was 24 h. Glycogen phosphorylase inhibitor (GPI; Sigma-Aldrich, St. Louis, MO, USA) treatment of RAW264.7 cells was timed at 30 min prior to LPS stimulation. Human monocytic THP-1 cells were also purchased from the Chinese Academy of Medical Science and cultured in complete RPMI-1640 medium (Servicebio, Wuhan, China) containing 10% FBS, 0.05 mM β-mercaptoethanol (Solarbio, Beijing, China) and 1% penicillin-streptomycin. THP-1 cells were differentiated into adherent macrophages by incubation with 100 ng/mL phorbol 12-myristate 13-acetate (PMA) (Sigma-Aldrich, St. Louis, MO, USA) for 72 h. Then, these cells were treated with the same conditions as for RAW264.7 macrophages to induce inflammation.

### Cell viability assay

RAW264.7 cells and differentiated THP-1 cells were treated with LPS or MK8617 for 24 h, then transferred to serum-free medium containing cell counting kit-8 (CCK-8) reagent

(APEXBIO, Houston, TX, USA) (10 μL of CCK-8 per 100 μL of medium) and incubated at 37 °C for 1 h. Cell viability was showed by measuring the absorbance at 450 nm.

## Real-time quantitative reverse-transcription polymerase chain reaction

Treated cells were collected for total RNA extraction with RNA Isolater Total RNA Extraction Reagent (Vazyme Biotech, Nanjing, China) and reverse transcribed using HiScript III RT SuperMix (Vazyme Biotech, Nanjing, China). qRT-PCR was performed using ChamQ Universal SYBR qPCR Master Mix (Vazyme Biotech, Nanjing, China). The specific reaction conditions are shown in Table S1. The primers (GentleGer, Suzhou, China) used for real-time fluorescent qPCR are listed in Table S2.

## Western blotting

Treated cells were collected by adding RIPA buffer (Beyotime, Shanghai, China), phenylmethylsulfonyl fluoride (Beyotime, Shanghai, China), and phosphatase inhibitor (Beyotime, Shanghai, China) for 30 min, and the supernatant was collected by centrifuging the lysate (12,000 rpm, 15 min). Protein samples were separated by 10% sodium dodecyl-sulfate polyacrylamide gel electrophoresis and transferred to polyvinylidene fluoride membranes, which were then closed at room temperature for 10 min with Rapid Closure Solution. The membranes were then incubated for 12 h using appropriate primary antibodies against the following proteins: inducible nitric oxide synthase (iNOS; 1:1,000; Cell Signaling Technology, Boston, MA, USA); CD80 (1:2,000; Proteintech, Wuhan, China); GYS1 (1:5,000; Abcam, Cambridge, MA, USA); HIF-1α (1:1,000; Proteintech, Wuhan, China); P2Y$_{14}$ (1:1,000; ABclonal, Wuhan, China) and β-actin (1:2,000; Proteintech, Wuhan, China). Next, secondary antibody:HRP-labeled goat anti-rabbit immunoglobulin G (1:1,000; Beyotime, Shanghai, China) was used at room temperature for 1 h. Protein signals were captured with a Gel Document System (SYNGENE, Cambridge, UK), and quantitative analysis of the protein was performed using ImageJ software.

## Enzyme-linked immunosorbent assay

Treated cell supernatants were collected to measure UDPG content in the supernatant with an ELISA kit (Enzyme-linked Biotech, Shanghai, China), according to the manufacturer's instructions, and then absorbance was measured at 450 nm.

## Immunofluorescence

RAW264.7 cells were seeded at a density of $2 \times 10^5$ cells/mL, then cultured and treated. Once 70–80% confluency was reached, RAW264.7 cells were fixed with 4% paraformaldehyde and blocked with Immunol Staining Blocking Buffer (Beyotime, Shanghai, China). Then, the cells were incubated with primary antibodies overnight at 4 °C. Primary antibodies for IF are included: anti-iNOS (1:500; CST, Danvers, MA, USA), anti-P2Y$_{14}$ (1:200; ABclonal, Wuhan, China). Next day, goat anti-rabbit secondary IgG antibody (1:500; Abcam, Cambridge, MA, USA) was used at room temperature for 1 h. Cell nuclei were counterstained with 4,6-damidinyl-2-phenylindole (DAPI; 5 μg/mL,
Beyotime, Shanghai, China), and the cells were imaged using a fluorescence microscope (Leica, Oscar, Germany).

### Lentiviral (LV) transfection of HIF-1α interference (HIF-1αi) and GYS1 interference (GYS1i)

The short hairpin (sh) RNAs targeting the HIF-1α and GYS1 gene promoters were designed and loaded into lentiviral by Genechem (Shanghai, China). shRNA sequences are listed in Table S3. Lentiviruses are used for infection at an MOI of 100. With RAW264.7 cells at approximately 40% confluence, the lentiviral-loaded shRNA-HIF-1α (LV-HIF-1αi) or lentiviral-loaded negative control (LV-NC) was added into DMEM medium containing 10% FBS for 12 h incubation at 37 °C; then, the cells were cultured in DMEM with 10% FBS for another 60 h after removing lentiviral. Transfection of HIF-1α was similar to that of GYS1. Transduction efficiency was assessed by the percentage of GFP-positive cells observed under an inverted fluorescence microscope.

### Statistical analysis

All data were from at least three independent experiments and expressed as mean ± standard error of the mean (SEM). GraphPad Prism 9 software was used to analyze data. Two groups of data were analyzed by t-test, and multiple groups were analyzed by one-way analysis of variance (ANOVA). Statistical significance was indicated by a $p$ value ≤ 0.05.

## RESULTS

### MK8617 pretreatment attenuates M1 macrophage inflammation

Before the formal experiment, we set the concentration gradient and mapped the target concentration range of MK8617 acting on RAW264.7 macrophages or THP-1 cells. The results showed that there was no significant cytotoxic effect after treating cells with MK8617 at concentrations <1 µM for 24 and 48 h, compared to vehicle-treated control cells (Figs. S1A, S1B). Therefore, we limited the concentration of MK8617 to <1 µM in the subsequent experiments. The duration of treatment was chosen as 24 h. LPS at a concentration of 1 µg/mL was used to induce macrophage polarization and inflammation, which had no effect on macrophage viability (Figs. S1C, S1D). In RAW264.7 cells, MK8617 pretreatment downregulated the M1 macrophage marker CD80 (Figs. 1A, 1B) and a variety of inflammatory markers, including tumor necrosis factor-α (TNF-α) (Fig. 1C), interleukin-1β (IL-1β) (Fig. 1D), interleukin-6 (IL-6) (Fig. 1E), and nitric oxide synthase (iNOS) (Figs. 1F–1H), that were elevated due to LPS treatment; as MK8617 concentration increased, the anti-inflammatory effect became more pronounced. And these indicators were not disturbed by MK8617 in normal cells (Fig. S2). Thus, MK8617 pretreatment reduced M1 macrophage inflammation in mice. To validate the above murine data in human macrophages, we additionally repeated the experiments in human monocytic THP-1 cells and obtained the same results (Fig. S3). In addition, we investigated whether post-treatment with MK8617 also suppressed inflammation. However, both in mouse macrophages (Figs. S4A–S4C) and human macrophages (Figs. S4D–S4F), LPS-induced inflammation was not reduced by the subsequent addition of MK8617. Therefore, it can be

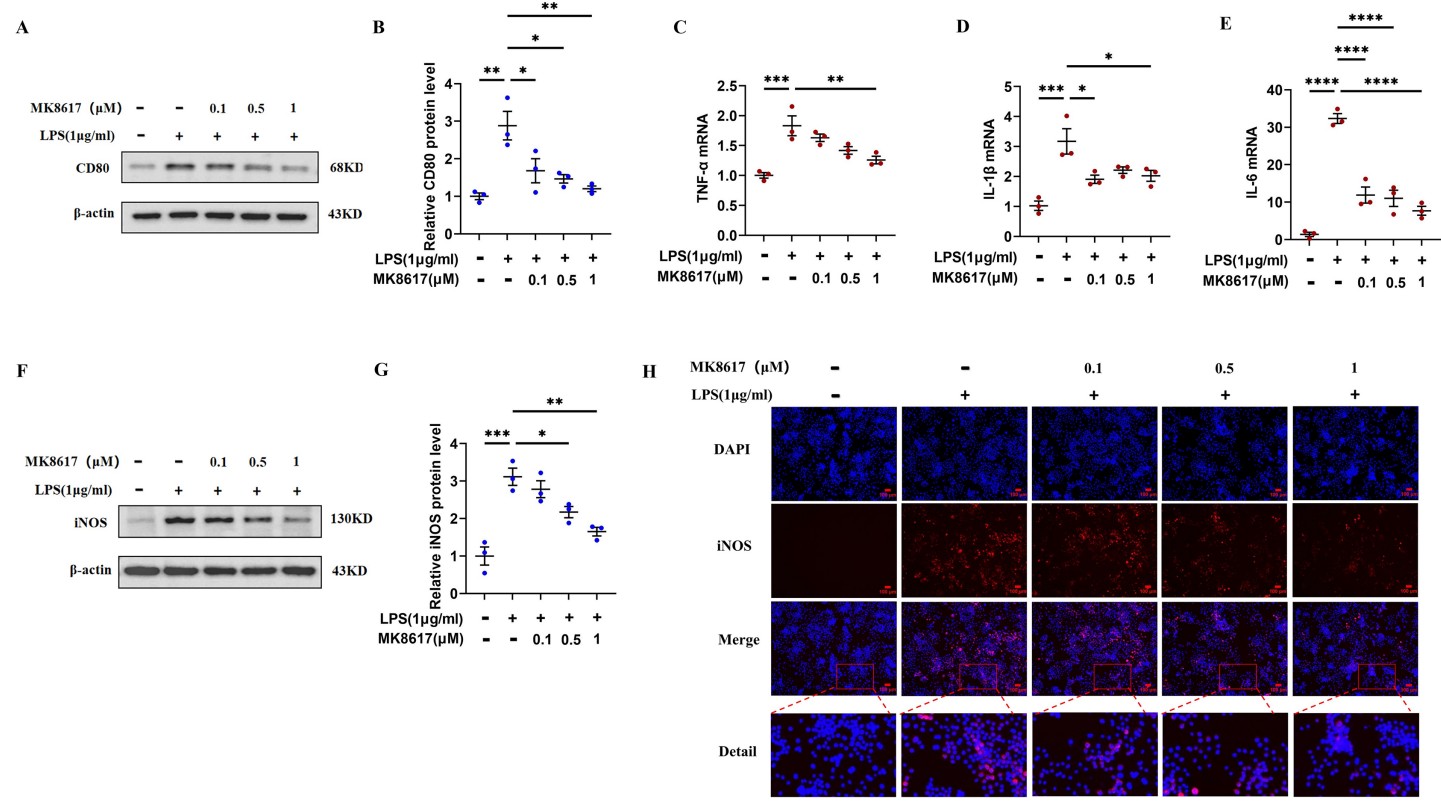

**Figure 1 MK8617 attenuates M1 macrophage inflammation.** (A) Western blotting result of CD80. (B) Quantification of the data in (A). mRNA levels were determined using RT-PCR: (C) TNF-α, (D) IL-β and (E) IL-6. (F) Western blotting result of iNOS. (G) Quantification of the data in (F). (H) iNOS expression was determined by immunofluorescence. Scale bars, 100 µm. Data are mean ± SEM ($n$ = 3). $^*p < 0.05$, $^{**}p < 0.01$, $^{***}p < 0.001$, $^{****}p < 0.0001$. "−" represents none and "+" represents yes.

concluded that MK8617 pretreatment attenuates M1 macrophage inflammation. As the 1 µM concentration of MK8617 had the most pronounced anti-inflammatory effect, this concentration was selected for further exploration.

## MK8617 inhibits the production of UDPG and P2Y$_{14}$ in M1 macrophages

We next explored how MK8617 exerted its anti-inflammatory effects. We measured UDPG content in cell supernatants by ELISA and showed that UDPG secretion was increased after LPS stimulation of macrophages, while MK8617 pretreatment reduced UDPG production (Fig. 2A). In addition, qRT-PCR results showed that P2Y$_{14}$ mRNA levels were correspondingly increased in M1 macrophages, whereas pretreatment with MK8617 downregulated P2Y$_{14}$ mRNA levels (Fig. 2B). Consistently, western blotting (Figs. 2C, 2D) and immunofluorescence (Fig. 2E) of P2Y$_{14}$ showed the same results. This suggests that MK8617 may exert anti-inflammatory effects by inhibiting UDPG and P2Y$_{14}$ production in M1 macrophages.

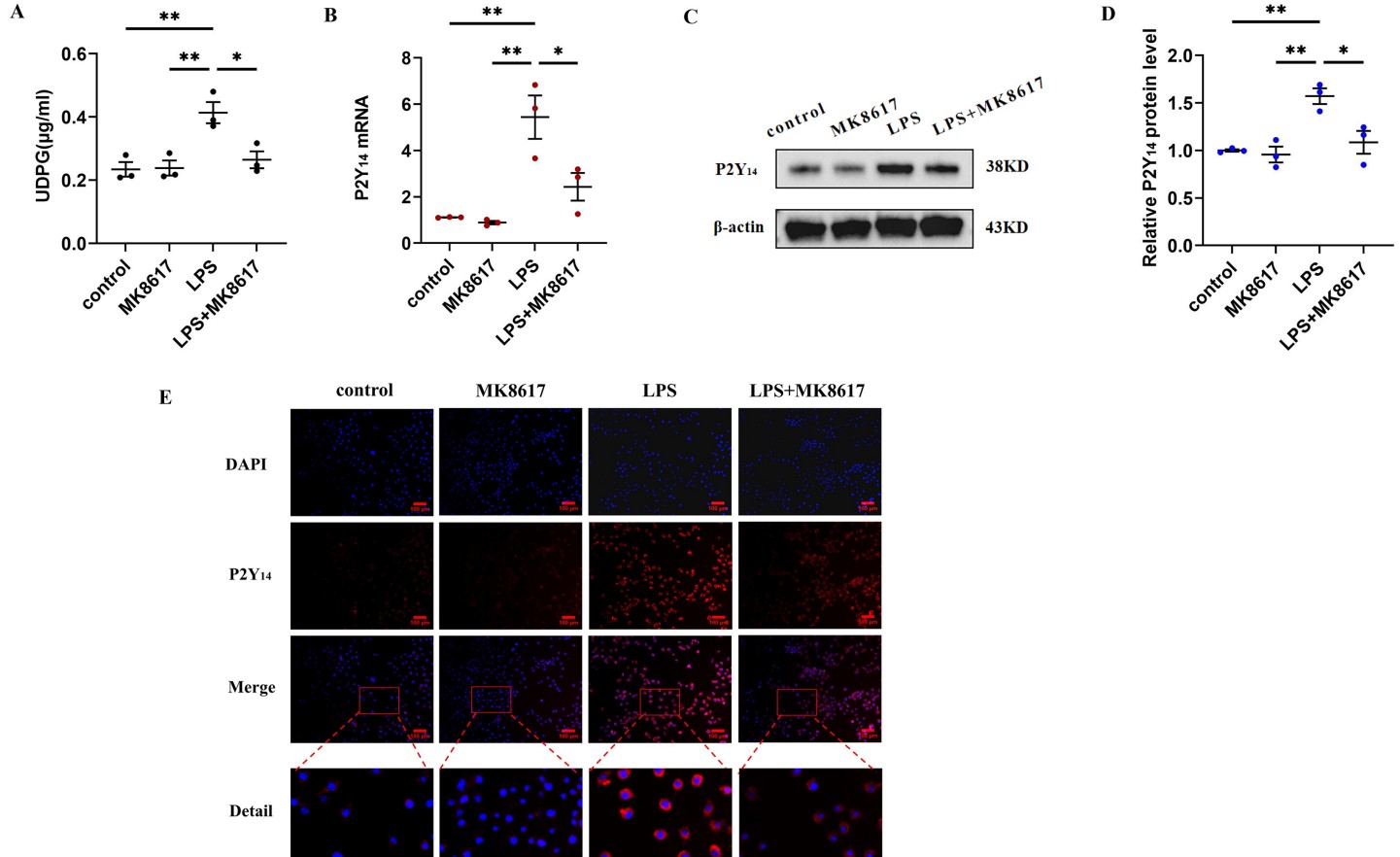

**Figure 2 MK8617 inhibits the production of UDPG and P2Y$_{14}$ in M1 macrophages.** (A) The supernatant was collected for determination of UDPG content by ELISA. (B) P2Y$_{14}$ mRNA level was determined using qRT-PCR. (C) Western blotting result of P2Y$_{14}$. (D) Quantification of the data in (C). (E) P2Y$_{14}$ expression was determined by IF. Scale bars, 100 μm. Data are mean ± SEM ($n = 3$). $^*p < 0.05$, $^{**}p < 0.01$.

## UDPG binding to P2Y$_{14}$ receptors induces macrophage polarization and inflammation

To investigate whether the anti-inflammatory effect of MK8617 was due to the inhibition of UDPG and P2Y$_{14}$ production in M1 macrophages, we stimulated RAW264.7 macrophages using different concentrations of UDPG (100, 200, 400 μM). UDPG upregulated P2Y$_{14}$ (Figs. 3A–3D), CD80 (Figs. 3E, 3F), and inflammatory indicators in macrophages (Figs. 3G–3L), and their production were higher with increasing UDPG concentrations. GPI inhibits glycogenolysis and reduces UDPG production (*Ma et al., 2020*). We inhibited UDPG in inflammatory macrophages with GPI (Fig. S5A), and then P2Y$_{14}$, CD80 and iNOS were subsequently downregulated (Figs. S5B–S5I). These results suggest that UDPG binds to P2Y$_{14}$ receptors to induce macrophage polarization and inflammation. Therefore, MK8617 inhibited production of UDPG and P2Y$_{14}$ in M1 macrophages, thereby reducing inflammation.

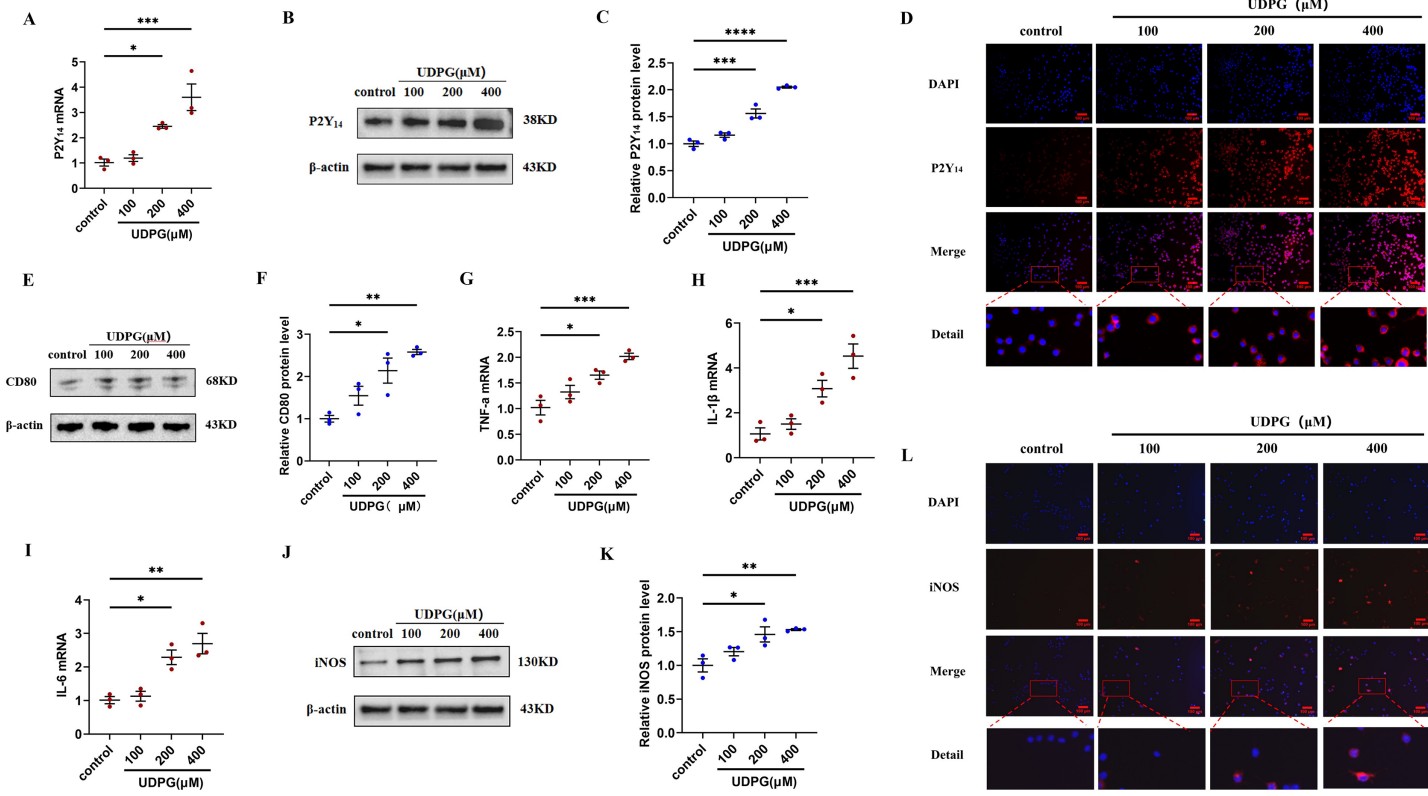

**Figure 3 UDPG binding to P2Y$_{14}$ receptors induces macrophage polarization and inflammation.** (A) mRNA expression of P2Y$_{14}$ by qRT-PCR. (B) Western blotting result of P2Y$_{14}$. (C) Quantification of the data in (B). (D) P2Y$_{14}$ expression was determined by IF. Scale bars, 100 μm. (E) Western blotting result of CD80. (F) Quantification of the data in (E). mRNA levels were determined by qRT-PCR: (G) TNF-α; (H) IL-1β and (I) IL-6. (J) Protein expression of iNOS by WB. (K) Quantification of the data in (J). (L) iNOS expression was determined by IF. Scale bars, 100 μm. Data are mean ± SEM ($n = 3$). $^*p < 0.05$, $^{**}p < 0.01$, $^{***}p < 0.001$, $^{****}p < 0.0001$.

## MK8617 reduces UDPG and P2Y$_{14}$ production by upregulating HIF-1α/GYS1

To deeply investigate the mechanism by which MK8617 inhibited the production of UDPG and P2Y$_{14}$, we compared the mRNA and protein expression of HIF-1α and GYS1 in four groups of macrophages: control macrophages, macrophages with LPS (M1) and MK8617 alone, and M1 macrophages pretreated with MK8617. The addition of MK8617 or LPS alone to macrophages increased HIF-1α and GYS1, both of which were further increased in M1 macrophages pretreated with MK8617 (Figs. 4A–4E). GYS1 can mediate the synthesis of glycogen by UDPG, so the above results suggest that MK8617 may reduce the production of UDPG and P2Y$_{14}$ by upregulating HIF-1α/GYS1. To verify this mechanism, we knocked down the HIF-1α and GYS1 genes in macrophages (Figs. S6A–S6F). Knockdown of HIF-1α was followed by a corresponding decrease in GYS1 (Figs. 4F–4H). In inflammatory macrophages without GYS1 knockdown (LPS+LV-NC), MK8617 inhibited UDPG secretion and P2Y$_{14}$ production, but in cells with GYS1 knockdown (LPS+LV-GYS1i), MK8617 failed to downregulate UDPG and P2Y$_{14}$ (Figs. 4I–4L). This suggests that knockdown of GYS1 disrupts the action of MK8617. Overall, MK8617 reduced UDPG and P2Y14 production through upregulation of HIF-1α/GYS1.

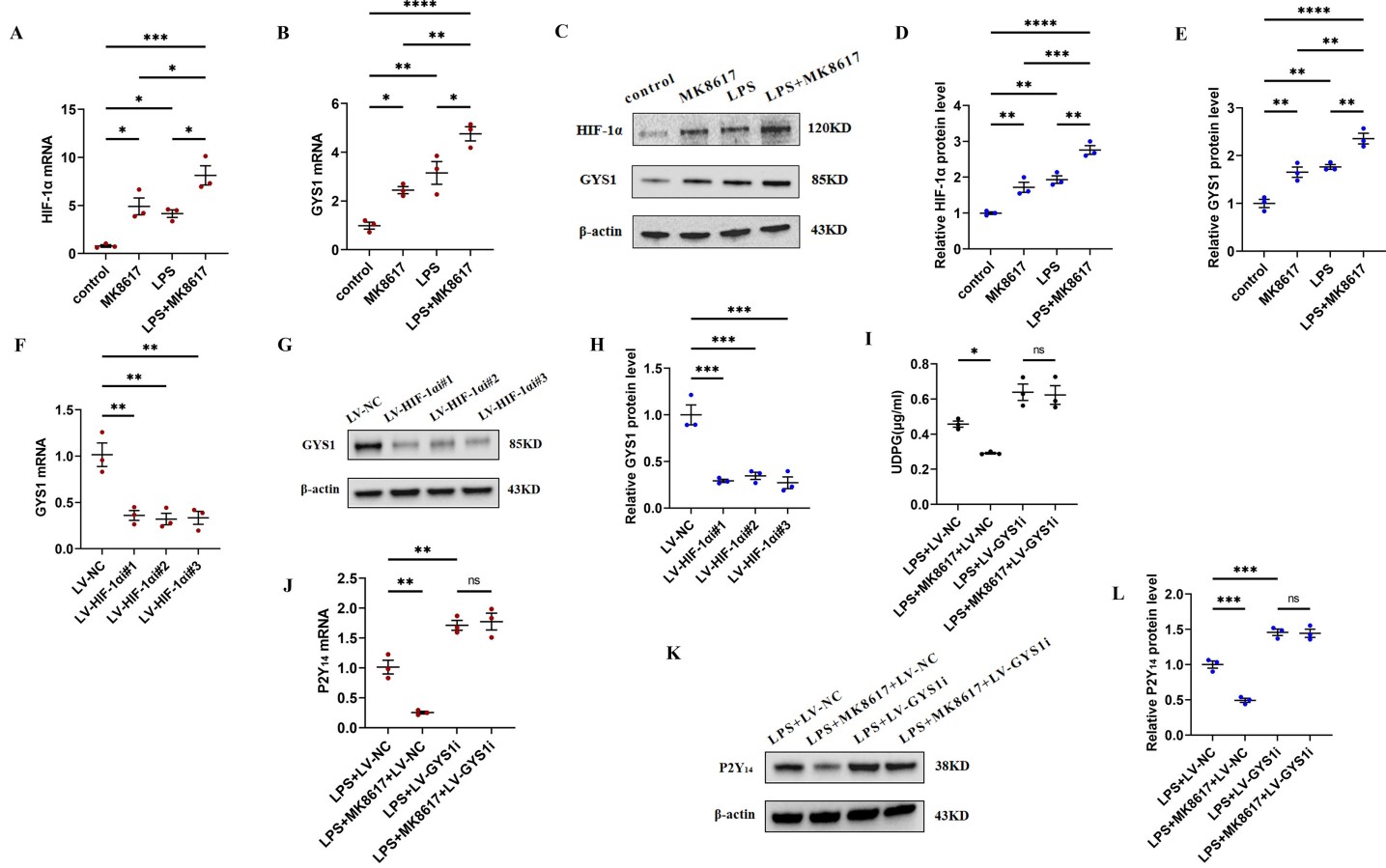

**Figure 4** **MK8617 reduces UDPG and P2Y$_{14}$ production through upregulation of HIF-1α/GYS1.** mRNA levels were determined by qRT-PCR: (A) HIF-1α (B) GYS1. (C–E) WB results: HIF-1α and GYS1. (F–H) qRT-PCR and WB determination of mRNA and protein levels of GYS1 after HIF-1α was knocked down. (I) Supernatant was collected to determine UDPG content by ELISA. (J–L) mRNA and protein levels of P2Y$_{14}$ were determined by RT-PCR and WB. Data are mean ± SEM ($n = 3$). $^*p < 0.05$, $^{**}p < 0.01$, $^{***}p < 0.001$, $^{****}p < 0.0001$.

## HIF-1α and GYS1 knockdown disrupt MK8617-induced suppression of inflammation

In inflammatory macrophages without knockdown of HIF-1α or GYS1 (LPS+LV-NC), MK8617 inhibited TNF-α, IL-1β, IL-6, and iNOS production, but in cells with HIF-1α or GYS1 knockdown (LPS+LV-HIF-1αi or LPS+LV-GYS1i), MK8617 failed to downregulate the above inflammatory indicators (Figs. 5A–5J). Therefore, HIF-1α and GYS1 have crucial roles for MK8617 to exert its anti-inflammatory effects. Overall, MK8617 modulated the HIF-1a/GYS1/UDPG/P2Y$_{14}$ pathway to attenuate M1 macrophage inflammation.

## DISCUSSION

In this study, we showed that safe doses of MK8617 significantly attenuated the inflammatory response in M1 macrophages. This effect was partly due to modulation of GYS1, by HIF-1α interfering with the binding of UDPG to P2Y$_{14}$ and the resulting release of inflammatory mediators (Fig. 6).

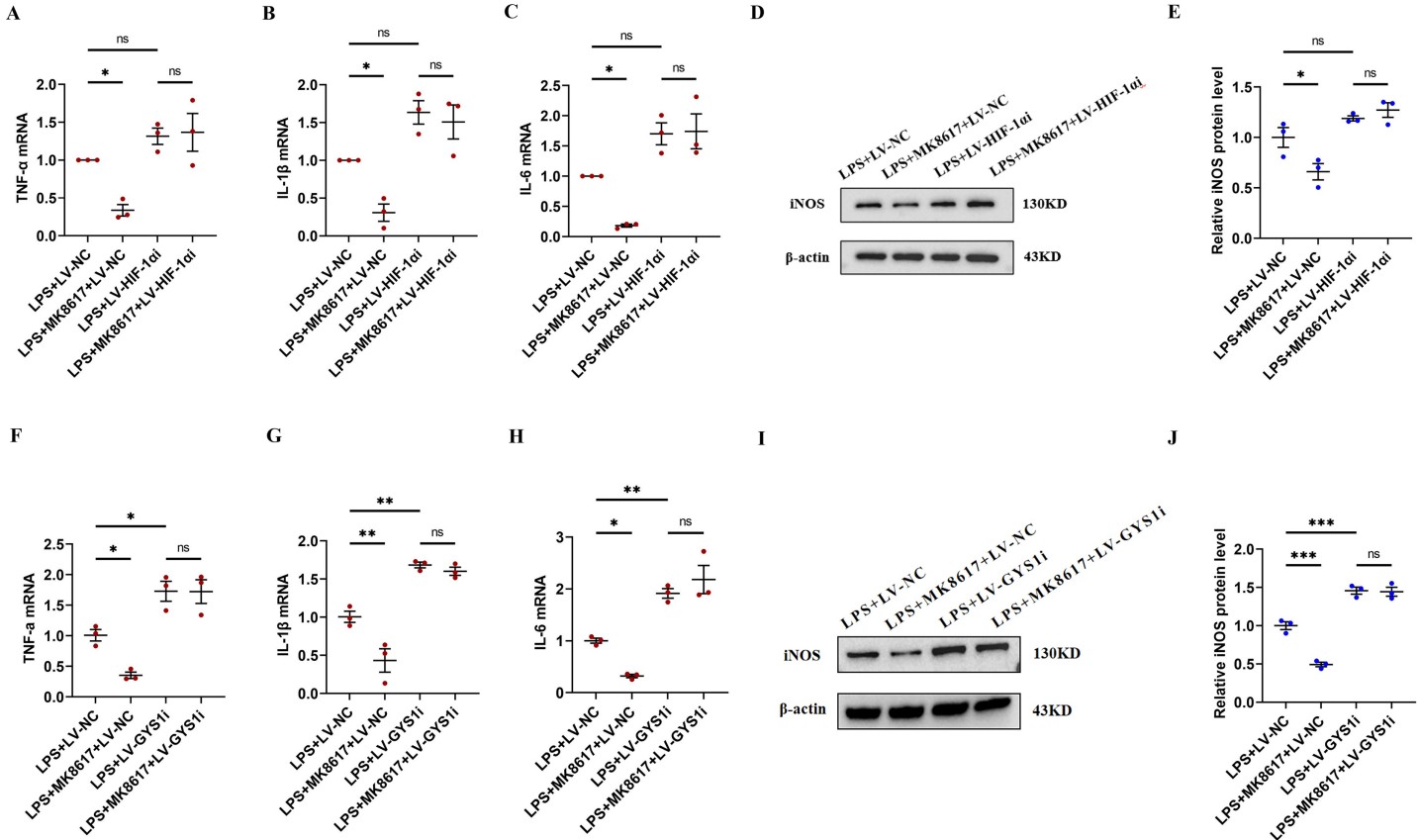

**Figure 5  HIF-1α and GYS1 knockdown disrupt MK8617-induced suppression of inflammation.** HIF-1α was knocked out in macrophages and then LPS was used to induce inflammation. mRNA levels of inflammatory cytokines were determined using qRT-PCR: (A) TNF-α (B) IL-β (C) IL-6 (D, E) WB was used to determine iNOS expression. GYS1 was knocked out in macrophages and then LPS was used to induce inflammation. mRNA levels of inflammatory cytokines were determined using qRT-PCR: (F) TNF-α (G) IL-β (H) IL-6 (I, J) WB was used to determine iNOS expression. Data are mean ± SEM ($n = 3$). *$p < 0.05$, **$p < 0.01$, ***$p < 0.001$.               

Recently, several studies reported that HIF-PHIs protect against various inflammation-associated diseases (*Miao et al., 2021*; *Han et al., 2020*; *Kim et al., 2021*). LPS is a common contributor to inflammatory responses, and it induces macrophages to release various pro-inflammatory factors, including TNF-α, IL-1β, IL-6, and iNOS; excessive release of these factors leads to extensive tissue damage and pathologic changes (*Orecchioni et al., 2019*; *Maldonado, Sá-Correia & Valvano, 2016*). Therefore, as we described previously, RAW 264.7 macrophages and THP-1 cells were treated with LPS to polarize to M1 phenotype, with the aim of exploring the anti-inflammatory activity and potential mechanisms of the novel HIF-PHI MK8617. Levels of the abovementioned inflammatory mediators were markedly increased in M1 macrophages, indicating that the inflammation model was successful. As expected, the highest concentration of MK8617 (1 μM) used in the study significantly inhibited release of these inflammatory mediators. The mapping of CCK-8 for target concentrations of MK8617 also showed that the MK8617 concentration used in our study was not toxic to macrophages. Therefore, the anti-inflammatory effect of MK8617 was not due to poor cell viability. Overall, unlike high doses of MK8617, safe doses of MK8617 have an inhibitory effect on inflammation. In our

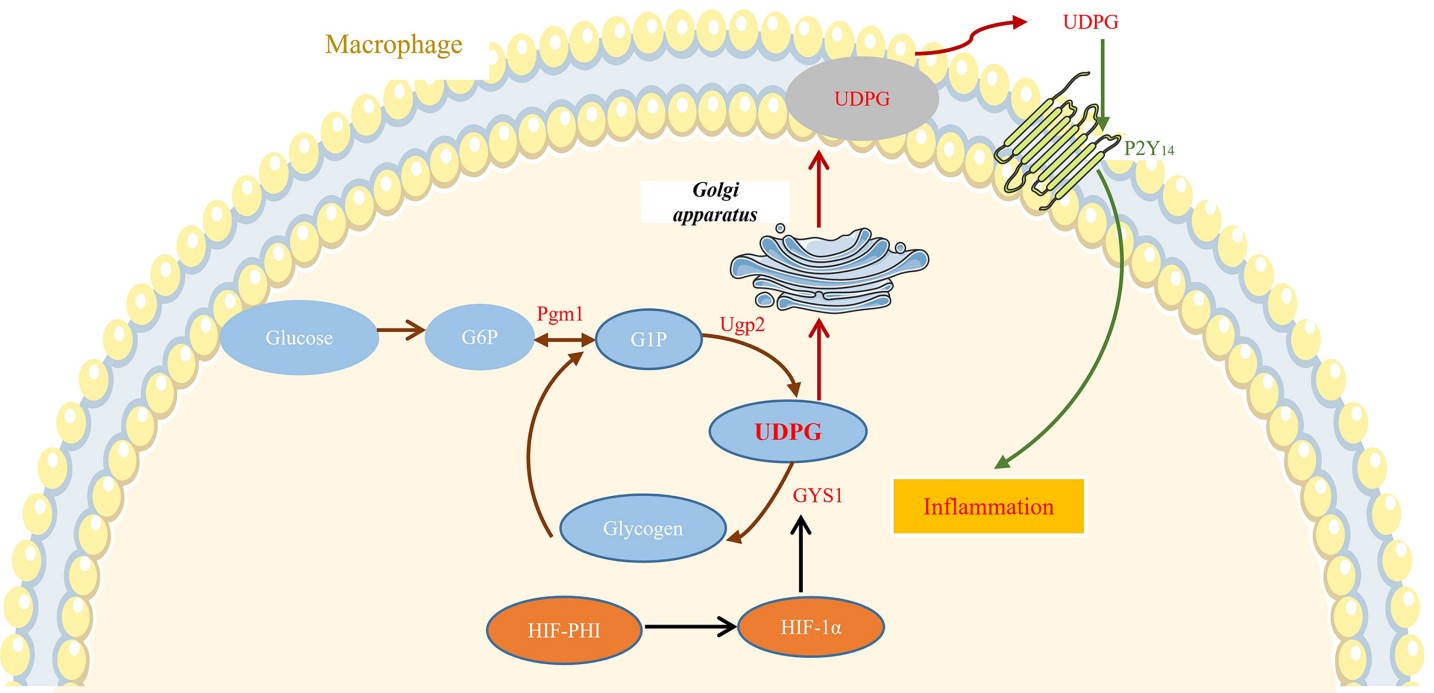

**Figure 6 MK8617 inhibits M1 macrophage inflammation *via* the HIF-1α/GYS1/UDPG/P2Y$_{14}$ pathway.** HIF-PHI upregulated GYS1 by stabilizing HIF-1α, and GYS1 was able to mediate glycogen synthesis by UDPG. Secretion of extracellular UDPG bound to P2Y$_{14}$ was thus reduced and the initiation of inflammation was inhibited.

previous study, MK8617 had a similar dose-dependent biphasic effect on renal tubulointerstitial fibrosis (TIF) (*Li et al., 2019*).

Previous studies reported a central function for UDPG in regulating the inflammatory response of macrophages through activation of inflammatory signaling by binding to P2Y$_{14}$ receptors (*Ma et al., 2020*; *Breton & Brown, 2018*). In this study, we attempted to verify the importance of the UDPG/P2Y$_{14}$ pathway in our model of inflammation. Earlier work from *Jokela et al. (2014)* showed UDPG: is a direct donor of glucose in the synthesis of glycogen; and belongs to the damage-associated molecular pattern, is released by damaged cells, and acts as a danger signal to induce onset of the immune response. Consistent with our study, the corresponding increase in cellular UDPG secretion after LPS stimulation provides some evidence that UDPG is a danger signal and that the early addition of MK8617 reduces UDPG secretion. Transduction of the next danger signal requires involvement of the purinergic G protein-coupled receptor P2Y$_{14}$, which is an inflammatory mediator present on the membranes of immune cells, including macrophages (*Klaver & Thurnher, 2021*). UDPG secreted extracellularly as a ligand can bind directly to P2Y$_{14}$, thus initiating a downstream inflammatory response (*Lazarowski & Harden, 2015*). Our results show that, in parallel with the amount of UDPG secreted, P2Y$_{14}$ receptors on cell membranes are increased by the addition of LPS, and such an increase in P2Y$_{14}$ is reduced by pretreatment with MK8617. Generally, these results are attributed to the change in UDPG secretion, because we have shown that exogenous

UDPG stimulation can also increase P2Y$_{14}$ expression, thus leading to macrophage-induced inflammation.

Intriguingly, GYS1, the enzyme that mediates the synthesis of glycogen by UDPG, appears to be directly regulated by HIF-1α (*Nuttall et al., 1994*; *Tiana et al., 2012*). In this study, MK8617 stabilized the expression of HIF-1α, and GYS1 increased with the increase of HIF-1α. Moreover, knockdown of the HIF-1α gene interfered with GYS1, both at the mRNA and protein level, which suggested that GYS1 was a downstream target gene of HIF-1α. Therefore, as a stabilizer of HIF, MK8617 can upregulate GYS1 to promote the synthesis of glycogen by UDPG and thus reduce the secretion of UDPG. It was shown that LPS stimulation of macrophages upregulated not only GYS1 but also phosphoglucomutase 1 (Pgm1) and UDP-glucose pyrophosphatase 2 (Ugp2), which affected UDPG production (*Ma et al., 2020*) (Fig. 6). Our study confirmed these results and that this may explain the elevated GYS1 but still increased UDPG after the addition of LPS (Figs. S7A, S7B). Futhermore, the addition of MK8617 alone in our experiments caused GYS1 to be upregulated without any change in UDPG as a result, which is a point to ponder. We conjecture that the cells after MK8617 addition remain in a stable state, unlike those destroyed by LPS. The original intracellular glycogen metabolism would adjust itself to reach a balance so that the secretion of UDPG remained, and the upregulation of Pgm1 somewhat verified our conjecture (Figs. S7C, S7D). Intracellular metabolism is intricate and requires more in-depth studies to explore. Anyway, GYS1 is an important link between MK8617 and UDPG, because the inhibition of UDPG secretion by MK8617 is disrupted with the knockdown of GYS1. Critically, knockdown of HIF-1α and GYS1 with lentivirus removed the preventive effect of MK8617 on inflammation. Therefore, *via* the HIF-1α/GYS1/UDPG/P2Y$_{14}$ pathway, MK8617 plays a role in controlling inflammation.

It has been previously reported that HIF-PHI pretreatment can have a protective effect on cells by adjusting glycogen metabolism (*Ito et al., 2020*). We also found that pretreatment of MK8617 adjusted intracellular glycogen metabolism in advance to the subsequent inflammatory environment by preventing the degradation of HIF, thus alleviating the imbalance of intracellular glycogen metabolism after LPS stimulation. The possible mechanism by which MK8617 pretreatment has an anti-inflammatory effect that post-treatment does not is the complexity of intracellular glycogen metabolism. Accordingly, in future applications of MK8617, we should pay attention to the timing of its use, as post-treatment with MK8617 was not able to reduce LPS-induced inflammation. The significant preventive effect of MK8617 on inflammation implied that it could potentially be applied as a prophylactic agent for inflammatory conditions, rather than as a therapeutic agent. Our research has value in terms of preventing inflammatory diseases.

## CONCLUSIONS

In conclusion, our study demonstrated the significant effect of safe doses of MK8617 in preventing inflammation and the specific regulatory mechanism: MK8617 reduced inflammatory signaling through the HIF-1α/GYS1/UDPG/P2Y$_{14}$ pathway, thereby attenuating M1 macrophage inflammation. These findings have, to some extent, revealed a direct association between glycogen metabolism and inflammation, providing new ideas

for the study of inflammation. The shortcoming of our research is the absence of an *in vivo* model to validate our ideas. Building on our foundation, future studies are needed to further determine the preventive effects and potential mechanisms of HIF-PHIs on inflammation in different *in vitro* and *in vivo* models, thus refining the understanding of HIF-PHIs and promoting their future application.

## ACKNOWLEDGEMENTS

We are thankful for the technology support from the Central Lab of Taizhou People's Hospital, China.

### Funding

The work was supported by Scientific Project of Taizhou (No. TS201901) and Taizhou People's Hospital Mandatory Project (No. ZL202012). The funders had no role in study design, data collection and analysis, decision to publish, or preparation of the manuscript.

### Grant Disclosures

The following grant information was disclosed by the authors:
The work was supported by Scientific Project of Taizhou: TS201901.
Taizhou People's Hospital Mandatory Project: ZL202012.

### Competing Interests

The authors declare that they have no competing interests.

### Author Contributions

- Lingling Qian conceived and designed the experiments, performed the experiments, analyzed the data, prepared figures and/or tables, authored or reviewed drafts of the article, and approved the final draft.
- Xiao-Qin Chen conceived and designed the experiments, analyzed the data, prepared figures and/or tables, and approved the final draft.
- Deyang Kong analyzed the data, prepared figures and/or tables, and approved the final draft.
- Gaoyuan Wang performed the experiments, prepared figures and/or tables, and approved the final draft.
- Yun Cao analyzed the data, prepared figures and/or tables, and approved the final draft.
- Yingchun Xiao conceived and designed the experiments, prepared figures and/or tables, and approved the final draft.
- Jing-Yuan Cao conceived and designed the experiments, authored or reviewed drafts of the article, and approved the final draft.
- Chunbo Zou conceived and designed the experiments, authored or reviewed drafts of the article, and approved the final draft.

## Data Availability

The raw data is available at figshare: Zou, Chunbo (2023). RAW DATA.zip. figshare. Dataset. https://doi.org/10.6084/m9.figshare.22494478.v1.

## Supplemental Information

Supplemental information for this article can be found online at http://dx.doi.org/10.7717/peerj.15591#supplemental-information.

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
