# Peer review of "MK8617 inhibits M1 macrophage polarization and inflammation via the HIF-1α/GYS1/UDPG/P2Y14 pathway"

_PeerJ, doi:10.7717/peerj.15591_

## Round 0.1 · original submission · Major Revisions

The manuscript has been assessed by three independent reviewers and I strongly suggest addressing the concerns raised by all three reviewers before your paper could be considered for publication.

1. Introduction has to be reframed with enough background and rationale for the study needs more justification.

2. Methods section needs elaboration, and the figures and legends need formatting. Authors are recommended to maintain uniformity in data representation.

3. It is suggested to discuss contradictory findings from previous studies (PMID: 31588798) and justify the reason behind the inconsistency in the results observed from the current study.

Reviewer 1 ·

Basic reporting

-

Experimental design

-

Validity of the findings

-

Additional comments

The present manuscript by Qian et al, explores the role of MK8617 in suppressing LPS induced inflammatory responses in RAW264.7 macrophages. Their findings indicate that MK8617 is able to inhibit M1 macrophage polarization and inflammation via HIF-1α/GYS1/UDPG/P2Y14 pathway.

Overall, the study is interesting but there are a few concerns that need to be addressed. First of all, the language used in abstract needs to be restructured, authors are requested to write full sentences for the statements of abstract. In the background part of abstract, you should mention that MK8617 is one of the HIF-PHI. Other concerns are listed below:

 Authors have shown here that pre-treatment with MK8617 attenuates LPS induced inflammation in RAW264.7 cells. If this finding is to be implied to clinical settings, it is important to also study if MK8617 is able to control the inflammation which is previously induced by LPS. Hence having MK8617 post-treatment group along with MK8617 pre-treatment group is needed to affirm its anti-inflammatory role.
These findings also need to be validated in human monocyte/ macrophage cell line as human and murine macrophages exhibit differential metabolic responses to LPS, and differ from each other in terms of their nitric oxide production.

 In figure panel 4, the results reveal that MK8617 alone or LPS+ MK8617 treatment induces the up-regulation of HIF-1α and GYS1. And you further discuss this finding stating that MK8617 reduces the production of UDPG and P2Y14 (Fig. 2) by up-regulating HIF-1α/GYS1. However, LPS alone treatment also leads to up-regulation of HIF-1α and GYS1 levels. How do you describe the increase of UDPG and P2Y14 levels in LPS alone group (Fig. 2) even after the induction of HIF-1α and GYS1?

 To further verify this mechanism, you performed lentiviral mediated silencing of HIF-1α and GYS1 in macrophages. However, the way this part has been described from line 199-205, it is very difficult to comprehend it. I would request authors to re-write this part in a way, that can allow easy understanding. Also, you need to follow alphabetical order for arranging images in figure panel.

 The title of manuscript indicates that MK8617 prevents M1 polarization, based on decrease of inflammatory cytokines and iNOS levels. However, it would be useful to assess the levels of M1 polarization markers such as pSTAT1, HLA-DR, CD80, CD86 and M2 polarization markers, CD206 and CD163.

 In the introduction section, first mention how inflammatory immune responses are initiated in the body (mention PRRs, TLRs, DAMPs etc. molecules associated with initiation of inflammatory cascade), then the central role of macrophages in inflammation and involvement of inflammation in various diseases, then stress upon how controlling macrophage induced inflammatory responses can be a key therapeutic strategy for autoimmune and inflammatory diseases.

 The methods should be described in details sufficient enough for other researchers to replicate your experiments once the manuscript is accepted for publication.
o For cell viability assay, RAW264.7 cells were treated and changed to serum free media before adding CCK-8 reagent. Give brief details of the timeline, as to when cells were transferred to serum free media and when CCK-8 was added to the culture? Line no. 95-99 is repetition of line no. 101-105.
o For qRT-PCR, the source of primer pairs used, PCR cycling conditions, and annealing temperatures of primer pairs should be mentioned.
o In WB experiment, what was the composition of lysis buffer used to lyse cells, the % of SDS-polyacrylamide gel used to run the protein samples, concentrations and incubation period of primary and secondary Abs? Complete the WB methodology by mentioning the blot development conditions.
o What was the density of cells seeded for immunofluorescence staining, and which fixative was used for fixing the cells, was it 4% paraformaldehyde or neutral buffered saline or some other reagent? The concentrations of primary and secondary antibodies and DAPI should be mentioned.
o Provide sufficient details for lentiviral transfection experiments.

 The figure legends need to be improved. Mention p values at the end of each figure legend as mentioning them after each graph/ image, breaks the flow of text. You can mention them collectively as (* p≤0.05, ** p≤0.01, *** p≤0.001 and so on). You need to arrange individual figures in the panel in alphabetical order in which they are labelled.

 In Fig.1 you should also include MK8617 alone group to assess the levels of cytokines and iNOS. Correct the labelling for Fig.1(F). For all immunofluorescence images, it is important to provide scale bars.

 Maintain uniform nomenclature in the manuscript and figure panels.

 Though GYS1 is downstream of HIF-1α, why HIF-1α knock-out group was not used to evaluate cytokine levels (Fig. 5)?

Reviewer 2 ·

Basic reporting

In the current manuscript, the authors have investigated the mechanisms involved in mediating the anti-inflammatory properties of HIF-PHIs. The results are intriguing, but I am concerned with the current version and the results.

The authors need to reframe the introduction. They should introduce HIF, its significance, and then HIF-PHIs. Also, the context for UDPG is ambiguous and needs reframing. The introduction does not provide enough context to the paper's rationale.

Experimental design

1. Line 157, please mention vehicle-treated control cells instead of untreated cells.
2. It is suggested that the authors mention vehicle control instead of 0um in the figure legends when stating statistical comparisons.
3. What was the cell viability post-treatment with LPS? It is worthwhile to include this as a supplementary information
4. Explain the group comparisons using *, #, and & in the bar graphs. It is suggested that the authors avoid such designation and mention the p values in the graph itself, indicating comparison groups.
5. Though the authors mention 100X magnification for all the IFs, the cells look too small. It is suggested that the authors mention scale bars instead of magnification for the IF images. Also, I would suggest the authors provide the metadata associated with the IF images to clarify the magnification details and the exposures for the images as raw data.
6. It is suggested that the authors show morphological changes or M1-specific antibody staining or protein levels to confirm the M1 phenotype other than the levels of inflammatory markers. It is also essential to show this for the results in Fig 3. as the authors claim that treatment with UDPG alone can drive inflammatory phenotype in the macrophages.
7. The level of UDPG and P2Y remains unchanged in the MK8617 treatment alone; however, GYS1 and HIF are increasing for the same. What is the explanation for this?
8. Correct the spelling of Golgi in the graphical abstract
9. It would be confirmatory if the authors tried chemical inhibition of UDPG in the cells upon LPS treatment and checked inflammation.
10. The supplementary file should be in editable format.

Validity of the findings

None

Additional comments

None

Reviewer 3 ·

Basic reporting

Qian et al. study the effects of MK-8617 in reducing macrophage mediated inflammation in cultured RAW264.7 cells and dissect the underlying mechanism. They used LPS to induce M1 macrophage polarization and induction of inflammatory cytokines.
The language used throughout the manuscript is not very easy to follow and understand. Also, there are many examples of typos or duplicates sentences. Highly recommend getting the manuscript proofread. Please see the examples below.
Line 49, Line 276
Line 95-105: Duplication of cell viability assay method
Line 235: “As early as Jokela TA showed” could be potentially replaced with “Earlier work from Jokela TA showed”
Rationale for the study needs a lot more justification. Authors suggest all inflammation is bad and thus should be repressed is not true. Inflammation is double edged sword and a necessary mechanism, but excessive inflammation can lead to damaging effects. As such there are myriad of compounds that reduce inflammation and solid rationale behind using this drug is missing in the introduction of the study.

Experimental design

Overall, the experiments are well designed to support the findings. Some suggestions to make the visualization better are below.
There should be consistency in making the graphs. For example, Fig 2A and 2B can both be made using either bar graph or scatter graph. It would be helpful for the reader if significance symbols are shown above the comparisons lines as shown in Fig 2A to remove any ambiguity.
Also, the use of different symbols to show significance throughout the figures is confusing. The legends in almost all the figures are filled with comparison p-values/symbols and the text seems hidden in between which makes reading difficult.

Validity of the findings

All the observations were made in the cultured cells and thus it is very difficult to interpret if this study will have any impact on the development of MK8617 as an anti-inflammatory agent. The authors need to replicate their findings in mice – at least the major findings that MK8617 can reduce macrophage mediated inflammation – to bolster their conclusion that “MK8617 is expected to be a promising future anti-inflammatory agent”.
Authors are suggested to review the research article (PMID: 31588798) where it has been shown that MK-8617 increases the levels of MCP-1, TNF-α and IL-6 in vivo in mice which directly contradicts the findings in the current study.
Overall, the importance of findings and their impact on future research is not very easily gauged from the manuscript.

---

## Round 0.2 · Minor Revisions

The authors are asked to include figure legends, experimental conditions and clinical application of MK8617 in the revised manuscript to be considered for publication.

Reviewer 1 ·

Basic reporting

I appreciate authors for performing the suggested experiments and revising their manuscript according to raised concerns. There are a few minor points that need to be addressed though:
• Line 52 – Introduction: change ‘infectious’ to ‘infections’
• Post-treatment with MK8617 was not able to reduce LPS induced inflammation. So how do you explain the importance of MK8617 as an inflammation prevention agent in terms of its future clinical application? Also, conditions for these supplementary studies should be provided.
• Where are the legends for supplementary figures?
• Legend for Fig. 1 (A) is missing.

Experimental design

-

Validity of the findings

-

Additional comments

-

Reviewer 2 ·

Basic reporting

The authors have adequately answered my initial concerns in the best possible way. The manuscript can be considered for publication.

Experimental design

None

Validity of the findings

None

Additional comments

None

Reviewer 3 ·

Basic reporting

The revised version of the manuscript has addressed majority of the concerns and is deemed suitable for publication.

Experimental design

no comment

Validity of the findings

no comment

---

## Round 0.3 · Minor Revisions

The authors have adequately addressed all the comments from the reviewers. MK8617 treatment did not reduce LPS induced inflammation, the authors need to discuss this aspect with respect to clinical implications in the manuscript. Figure legends have to be provided.

---

## Round 0.4 · accepted · Accept

The authors have adequately addressed all the comments and the manuscript is ready for acceptance.